# Substituted Nano-Hydroxyapatite Toothpastes Reduce Biofilm Formation on Enamel and Resin-Based Composite Surfaces

**DOI:** 10.3390/jfb11020036

**Published:** 2020-06-01

**Authors:** Andrei C. Ionescu, Gloria Cazzaniga, Marco Ottobelli, Franklin Garcia-Godoy, Eugenio Brambilla

**Affiliations:** 1Oral Microbiology and Biomaterials Laboratory, Department of Biomedical, Surgical and Dental Sciences, University of Milan, via Pascal 36, 20133 Milan, Italy; gloria.cazzaniga@yahoo.it (G.C.); marco.ottobelli@unimi.it (M.O.); eugenio.brambilla@unimi.it (E.B.); 2Bioscience Research Center and Clinical Research, College of Dentistry, University of Tennessee Health Science Center, 875 Union Avenue, Memphis, TN 38163, USA; fgarciagodoy@gmail.com

**Keywords:** fluoride(s), biofilm(s), *Streptococcus mutans*, bioreactor(s), enamel, composite materials, biomaterials, hydroxyapatite, nanostructured materials, dental

## Abstract

*Background*: Toothpastes containing nano-hydroxyapatite (n-HAp) substituted with metal ions provide calcium and phosphate ions to dental hard tissues, reducing demineralization, and promoting remineralization. Few data are available about the effect of these bioactive compounds on oral microbiota. *Methods*: This in vitro study evaluated the influence of two commercially-available substituted n-HAp-based toothpastes (α: Zn-carbonate substituted n-HAp; β: F, Mg, Sr-carbonate substituted n-HAp) on early colonization (EC, 12 h) and biofilm formation (BF, 24 h) by oral microbiota. Controls were brushed with distilled water. Artificial oral microcosm and *Streptococcus mutans* biofilms were developed using human enamel and a resin-based composite (RBC) as adherence surfaces. Two test setups, a shaking multiwell plate and a modified drip-flow reactor (MDFR), were used to simulate clinical conditions during the night (low salivary flow and clearance) and daytime, respectively. Energy-dispersive X-ray spectrometry (EDS) was used to evaluate specimens’ surfaces after toothpaste treatment. Fluoride release from β toothpaste was evaluated. Viable adherent biomass was quantified by MTT assay, and biofilms’ morphology was highlighted using confocal microscopy. *Results*: EDS showed the presence of remnants from the tested toothpastes on both adherence surfaces. β toothpaste showed significantly lower EC and BF compared to control using the artificial oral microcosm model, while α toothpaste showed lower EC and BF compared to control, but higher EC and BF compared to β toothpaste. The effect shown by β toothpaste was, to a minimal extent, due to fluoride release. Interestingly, this result was seen on both adherence surfaces, meaning that the tested toothpastes significantly influenced EC and BF even on RBC surfaces. Furthermore, the effect of toothpaste treatments was higher after 12 h than 24 h, suggesting that toothbrushing twice a day is more effective than brushing once. *Conclusions*: The efficacy of these treatments in reducing microbial colonization of RBC surfaces may represent a promising possibility in the prevention of secondary caries.

## 1. Introduction

Biofilms colonizing dental surfaces represent the habitat of a wide range of microorganisms. In healthy conditions, this microbial community provides fundamental benefits to host surfaces, including homeostasis of critical environmental parameters such as pH, as well as protection against colonization by exogenous species. However, loss of this equilibrium can result in dysbiosis, leading to the onset of oral diseases, including dental caries [1,2].

Mechanical biofilm control can be beneficially associated with the use of active compounds to reduce biofilm formation, positively affecting the biological equilibrium of the ecosystem [3]. The search for new bioactive compounds to prevent dental caries development and progression has led researchers to focus their attention on the use of nanotechnologies, especially hydroxyapatite (HAp), metals, and metal oxide nanoparticles [4,5,6]. Nanotechnology applications offer the opportunity to modulate the formation of dental biofilms using nanoparticles with bioactive effects. Indeed, their possible use in biofilm control is related to some peculiar characteristics that materials acquire when nano-sized: shape, surface-to-volume ratio, chemical reactivity, and ability to interact with the bacterial cell wall directly [6,7,8]. Considerable efforts have been made to clarify the transfer mechanisms of such nanoparticles through biofilms, indicating that nanoparticles can diffuse through water channels inside biofilms, interacting both with the microbial cells and with the hard tissue surface.

Toothpastes containing nano-hydroxyapatite (n-HAp) and its precursors can provide calcium and phosphate ions to dental hard tissues, reducing demineralization, and promoting remineralization. N-HAp may also form a protective layer on dental hard tissues with chemical characteristics very close to natural tissues, showing biomimetic properties [9,10]. Biomimetic n-HAp or other calcium phosphate-based nanosystems have been shown to actively shift the ionic balance toward remineralization [11,12]. Recently, n-HAp-based toothpastes containing metal ions such as Zn, F, Mg, and Sr have been introduced to improve the remineralization processes. The small quantity of metal ions with which nHAp is doped modulates the properties of the compound already from its deposition, influencing crystal growth and length, and influencing its solubility and strength. They are firmly embedded inside the lattice and are not supposed to be released. Most studies, therefore, provide data about the remineralization potential of these compounds on surface lesions, with few reporting their influence on early microbial colonization (EC) and biofilm formation (BF) [13,14,15,16].

Resin-based composites (RBCs) provide an example of artificial surfaces intensely colonized by cariogenic biofilms due to their lack of buffering effect [17]. This characteristic has been put into relation with the high occurrence of secondary caries in the sound dental tissues surrounding an RBC restoration [17]. The possibility of influencing microbial colonization of RBCs, therefore, can be seen as a means to prevent the occurrence of such lesions. For this reason, RBCs are a potential target of preventive measures exploiting the potential of nanotechnologies in order to reduce secondary caries occurrence.

The aim of this study was to evaluate the in vitro effect of two n-Hap-based toothpastes on early colonization (EC) (12 h) and biofilm formation (BF) (24 h) of human enamel and RBC surfaces. The study was performed by aerobically culturing the monospecies *Streptococcus mutans* and an artificial oral microcosm using both a static and a continuous flow setup. The aim of the two models was to assess the effect of the tested toothpastes under environments simulating clinical conditions during the night (low salivary flow and clearance) and day (high salivary flow and clearance). The null hypothesis was that toothpaste treatment would not influence EC and BF on sound enamel surfaces in the tested models. An additional null hypothesis was that toothpaste treatment would not influence EC and BF on RBC surfaces in the tested models.

## 2. Materials and Methods

### 2.1. Preparation of Specimens

Reagents, culture media, and disposables used in this study were obtained from E. Merck AG (Darmstadt, Germany). A total of 110 human teeth extracted for clinical reasons were obtained and stored at −20 °C until use. The Institutional Review Board of the University of Milan approved the protocol (codename: SALTiBO-2017). Enamel disks 6.0 mm in diameter and 1.5 mm thick were cut from the teeth surfaces by means of custom-made water-cooled diamond trephine burs (Indiam, Carrara, Italy). A total of 213 enamel disks were prepared for the study, using this procedure to obtain more than one disk from each tooth.

A total of 213 disks (6.0 mm diameter and 1.5 mm thickness) of a nanohybrid resin-based composite (RBC, Clearfil Majesty ES-2, shade A2, Kuraray Europe GmbH, Hattersheim am Main, Germany) were obtained by packing an excess of uncured material into a custom-made PTFE mold. The top and bottom surfaces of the RBC were covered with a cellulose acetate strip (Mylar) and condensed against a glass plate by applying a load of 1 kg for 20 s. The specimens were then irradiated for 40 s by placing the tip of a light-curing unit (MiniLED, Satelec, Acteon Group, Merignac, France, 420–480 nm emission, 1250 mW/cm^2^ irradiance) into direct contact with the acetate strip. To remove most of leachates, RBC specimens were then stored under light-proof conditions in phosphate-buffered saline (PBS) for 6 days at 37 °C, with the buffer replaced twice a day.

Both enamel and RBC disks were polished using grinding paper (1000, 2000, and 4000 grit) to obtain a surface roughness with Ra < 0.2 µm. They were then randomly divided into three groups (α, β, and control, n = 71/group, Figure 1).

### 2.2. Toothpastes

Two toothpastes containing n-HAp were tested (Table 1). The toothpastes were inserted into test tubes coded by the Greek letter corresponding to the first two groups (α, β). In this way, the experimenters performing microbiological procedures were blinded regarding their composition. Each toothpaste was mixed to a slurry using one part of toothpaste and two parts of distilled water. Then, enamel and RBC disks of groups α and β were manually brushed by a single operator with the corresponding toothpaste for 2 min and rinsed with distilled water for 1 min. Disks belonging to the control group were brushed with distilled water for 2 min, then rinsed for 1 min with distilled water. All disks were then sterilized using a chemical peroxide-ion plasma sterilizer (STERRAD, ASP, Irvine, CA, USA).

### 2.3. Surface Imaging and Analysis

Scanning electron microscopy (SEM) and energy-dispersive X-ray spectroscopy (EDS) analysis were performed (n = 4/group, Figure 1) using a TM4000Plus Tabletop scanning electron microscope (Hitachi, Schaumburg, IL, USA) equipped with an EDS probe (Q75, Bruker, Berlin, Germany) to investigate the presence of toothpaste residues on the specimen surface. Dry specimens were observed in surface-charge reduction mode without sputter-coating, using an accelerating voltage of 15 KV. Three randomly selected fields were acquired for each specimen at 500× magnification and were analyzed using the EDS probe in full-frame mode using an acquisition time of 150 s. SEM micrographs (backscattered electrons mode) and EDS elemental maps of the surfaces at 5000× magnification were also obtained. The acquired data represent the elemental composition of the ≈1 μm superficial layer.

### 2.4. Saliva Collection

Whole saliva was collected from five healthy volunteers, according to a previously described protocol [18]. They refrained from oral hygiene for 24 h, did not have any active dental disease, and did not have antibiotic therapy for at least three months prior to the experiment. Chilled test tubes were used for saliva collection. Saliva was then pooled, heated to 60 °C for 30 min and centrifuged (27,000× *g*, 4 °C, 30 min). The sterile supernatant was collected into sterile tubes and stored at −20 °C. Saliva was thawed at 37 °C for 1 h before use. The Institutional Review Board of the University of Milan approved the use of saliva (codename: SALTiBO-2017), and written, informed consent was obtained from all volunteers.

### 2.5. Microbiological Procedures

Two microbiological models were used in this study: a monospecific *S. mutans* culture and an artificial oral microcosm based on mixed oral flora (Figure 1).

A pure suspension of *S. mutans* strain ATCC 35668 was obtained as described elsewhere [19]. Briefly, Mitis Salivarius Bacitracin agar (MSB agar) plates were inoculated with the *S. mutans* strain and incubated aerobically in a 5% CO_2_-supplemented atmosphere at 37 °C for 48 h. After transferring an inoculum in Brain Heart Infusion (BHI) broth and further incubating in a 5% CO_2_-supplemented atmosphere at 37 °C for 12 h, a pure culture of the microorganism was obtained. Cells were harvested by centrifugation (2200× *g*, 19 °C, 5 min), washed twice with sterile phosphate-buffered saline (PBS), and resuspended in the same buffer., The suspension was sonicated using low-energy output (7W for 30 s, B-150 Sonifier, Branson, Danbury, CT, USA) to disperse bacterial chains. Finally, the suspension was adjusted to an optical density of 1.0 on the McFarland scale, corresponding to a concentration of approximately 6.0 × 10^8^ cells/mL.

A mixed oral flora inoculum was obtained from fresh, pooled saliva expectorated from the same five donors previously described, filtered through sterile glass wool, vigorously stirred for 2 min, and immediately used. Two test setups were used in this study: a shaking multiwell plate and a modified drip-flow reactor (MDFR).

#### 2.5.1. Shaking Multiwell Plate

Disks were placed in 48-well sterile plates and incubated at 37 °C for 24 h in sterile saliva; then, the saliva was removed by gentle aspiration. Each well was inoculated with 100 µl of either *S. mutans* suspension or mixed oral flora, and 900 µl of sterile modified artificial saliva medium. The medium composition was the following: 10.0 g/L sucrose, 2.5 g/L mucin (type II, porcine gastric), 2.0 g/L bacteriological peptone, 2.0 g/L tryptone, 1.0 g/L yeast extract, 0.35 g/L NaCl, 0.2 g/L KCl, 0.2 g/L CaCl_2_, 0.1 g/L cysteine hydrochloride, 0.001 g/L hemin, and 0.0002 g/L vitamin K1 [20]. Plates were then incubated aerobically in an orbital shaker at 37 °C and 100 rpm. After either 12 h (colonization) or 24 h (biofilm formation), adherent viable biomass assessment was performed.

#### 2.5.2. MDFR

The bioreactor used in the present study was a modification of a commercially available drip-flow reactor (DFR 110, BioSurface Technologies; Bozeman, MT, USA) according to previously described [21]. The modified design (MDFR) allowed the placement of customized PTFE trays on the bottom of the flow cells to submerge the specimen surfaces in the flowing medium. Specimens from each group were randomly divided into the eight flow cells of two identical MDFRs. After sterilization (STERRAD), the MDFR was assembled inside a sterile hood, and a salivary pellicle was obtained on the specimen surface after incubation with sterile saliva at 37 °C for 24 h. Saliva was then removed, each flow-cell was inoculated with 10 mL of either *S. mutans* suspension or mixed oral flora suspension to allow bacterial adherence. After 4 h, a peristaltic pump (RP-1k; Rainin, Emeryville, CA, USA) provided the constant flow of sterile modified artificial saliva medium through the flow cells (9.6 mL/h). For each microbiological inoculum, the amount of viable adherent biomass was evaluated after either 12 h or 24 h of further incubation in aerobic conditions (Figure 1).

### 2.6. Viable Biomass Assessment

Viable and metabolically active biomass adherent to the specimen surface was assessed using a tetrazolium-based assay as described previously [21]. In brief, a tetrazolium salt stock solution was prepared by dissolving 5 mg/mL 3-(4,5)-dimethylthiazol-2-yl-2,5-diphenyltetrazolium bromide (MTT) in sterile PBS; a phenazinium salt stock solution was prepared by dissolving 0.3 mg/mL of N-methylphenazinium methyl sulphate (PMS) in sterile PBS. The solutions were stored at 2 °C in light-proof vials until the day of the experiment when a fresh measurement solution (FMS) was prepared by diluting 1:10 v/v of MTT stock solution and 1:10 v/v of PMS stock solution in sterile PBS. A lysing solution (LS) was prepared by dissolving 10% v/v of sodium dodecyl sulphate (SDS) and 50% v/v dimethylformamide in distilled water and stored at 2 °C until the day of the experiment when it was warmed at 37 °C for 2 h before use. After the specified incubation times, the orbital shaker and the MDFR medium flow was halted, flow cells and plates were opened, the specimens were carefully removed and immediately placed into Petri plates containing sterile PBS at 37 °C. They were gently washed three times with sterile PBS to remove non-adherent cells and finally placed inside the wells of 48-well plates containing 300 μL of FMS each. The plates were incubated at 37 °C under light-proof conditions for 3 h. During incubation, electron transport across the microbial plasma membrane and, to a lesser extent, microbial redox systems, converted the yellow salt to insoluble purple formazan crystals. The conversion at the cell membrane level was facilitated by the intermediate electron acceptor (PMS). The unreacted FMS was gently removed by aspiration, and the formazan crystals were dissolved by adding 300 µL of LS to each well. The plates were stored for an additional 1 h under light-proof conditions at room temperature; 100 µL of the solution was then transferred into 96-well plates. The absorbance of the solution was measured using a spectrophotometer (Genesys 10-S, Thermo Spectronic, Rochester, NY, USA) at a wavelength of 550 nm; results were expressed as relative absorbance in optical density (OD) units corresponding to the amount of adherent, viable and metabolically active biomass.

### 2.7. Confocal Laser-Scanning Microscopy (CLSM)

A total of three specimens for each material and treatment group (Figure 1) were prepared for CLSM analysis. They were incubated using the MDFR test setup with the artificial oral microcosm model for 12 h. Then, they were gently removed from the flow cells, rinsed twice with sterile PBS, stained using the FilmTracer™ LIVE/DEAD^®^ Biofilm Viability Kit (Invitrogen Ltd., Paisley, UK), and analyzed using confocal laser-scanning microscopy (CLSM; Eclipse Ti2 inverted CLSM, Nikon, Tokyo, Japan). Three randomly selected image stack sections were recorded for each specimen. Confocal images were obtained using a dry objective (20×; NA = 0.5) at a resolution of 2048 × 2048 pixels, with a zoom factor of 1.0 and a scan speed of 400 Hz, and digitalized using the Nikon Imaging Software (NIS)—Elements Viewer, v. 4.50. Three channels were acquired in parallel scanning; the first used an excitation at 405 nm and emission at 420–470 nm in order to subtract potential autofluorescence digitally. The other two channels had an excitation wavelength of 488 nm, and emission was acquired at 500–570 nm (green channel, live bacteria) and 610–760 nm (red channel, dead bacteria). For each image stack section, 3D-rendering reconstructions were obtained using Drishti (Ajay Limaye, Australian National University, CAN, AUS, http://sf.anu.edu.au/Vizlab/drishti/).

### 2.8. Fluoride Release Measurements

Measurements of fluoride release from the fluoride-hydroxyapatite-containing toothpaste (β) were performed to correlate it with the hypothesized antimicrobial activity of the toothpaste. A total of 48 enamel disks and 48 RBC disks were additionally prepared and brushed as previously specified (cf. Section 2.1 and Section 2.2): one half (n = 24) was brushed with slurry from toothpaste β, and the other half was brushed with distilled water. Immediate fluoride release (t = 0) was evaluated from eight disks for each toothpaste. These disks were placed in 48-well plates containing 300 µL of sterile saliva in each well for 24 h. Disks were then discarded, 1:50 dilution of total ionic strength solution adjustment buffer, TISAB III (Orion Research Inc, Boston, MA, USA) was added, and the solution adjusted to pH = 5.5. The fluoride content of the solution was determined using the ion-selective electrode micro-method. The remaining disks were weighted with an analytical balance, then sterilized, and a monospecies *S. mutans* biofilm was allowed to develop on their surfaces after salivary pellicle formation using the MDFR as described in Section 2.5.2 for the same incubation times (12 h and 24 h). A total of eight disks for each toothpaste and time group were weighed again, and the weight of the adherent wet biomass was calculated. After that, each disk was inserted into one well of 48-well plates containing 300 µL of 10 vol% SDS in distilled water and 1:50 TISAB III (pH = 5.5), and the plates were sonicated for 5 min to allow dispersion of the biomass. Then, disks were discarded, and the fluoride content of the solution was determined.

The amount of released fluoride was calculated and displayed in parts per billion (ppb) after normalizing the fluoride readings by the disks’ surfaces. The amount of fluoride incorporated in the biofilm structures was calculated and displayed in parts per million (ppm) after normalizing the fluoride readings by the amount of adherent biomass. All analyses were performed using the ion-selective electrode micro-method, as described previously [22]. In brief, a stock solution with a fluoride concentration of 1000 ppm was appropriately diluted with 10 vol% SDS in distilled water at pH = 5.5 to obtain fluoride standards with fluoride concentrations ranging from 0.0019 to 64 ppm. A calibration curve was obtained using a digital pH/mV meter (SA-720, Orion Research Inc, Boston, MA, USA). A 1:50 final dilution of TISAB III as an ionic strength adjustor was added to each standard before the analyses. A negative reference standard (0 ppm fluoride) was prepared by adding 1:50 TISAB III to the solution containing 10 vol% SDS in distilled water; this solution was also used to rinse the electrodes between measurements.

### 2.9. Statistical Analysis

All statistical analyses were performed using statistical software (JMP 12.0, SAS Institute, Inc., Cary, NC, USA). The OD and F data were reported as means and standard errors calculated from the natural values. The normality of distributions was preliminarily checked using Shapiro-Wilk test, and homogeneity of variances was preliminarily checked using Bartlett’s test. A multi-way ANOVA model was used on OD dataset considering the factors: toothpaste (α, β, control), adherence surface (enamel, RBC), test setup (shaking multiwell plate, MDFR), microbiological model (*S. mutans*, oral microcosm), and incubation time (12 h, 24 h). A multi-way ANOVA model was used on fluoride dataset considering the factors: toothpaste (β, control), adherence surface (enamel, RBC), and incubation time (12 h, 24 h). Student-Newman-Keuls post-hoc test was used to highlight significant differences (*p* < 0.05).

## 3. Results

### 3.1. Surface Imaging and Analysis

To investigate the presence of toothpaste residues on enamel and RBC surfaces as well as the composition of the tested toothpastes, specimens were observed using SEM-EDS in surface-charge reduction mode without sputter-coating, using an accelerating voltage of 15 KV (Figure 2). EDS results showed that the elemental composition of both toothpastes was very similar, except that α toothpaste contained zinc, while β toothpaste contained strontium, magnesium, and fluoride (Figure 3). Enamel specimens showed a significant increase in surface zinc content after treatment with α toothpaste, and a significant increase in strontium and magnesium content after treatment with β toothpaste (Table 2). Very interestingly, EDS detected the presence of calcium and phosphate on the surfaces of RBC specimens treated with both toothpastes, as well as traces of zinc on specimens treated with α toothpaste and magnesium on specimens treated with β toothpaste (Figure 2, Table 2).

### 3.2. Microbiological Results

The effect of the tested toothpastes on EC and BF, expressed as mean OD values ± 1 standard error, according to test setup are shown in Figure 4 and Figure 5. ANOVA results showed an overall significant influence of the tested toothpastes on both EC and BF, with a significant interaction between the factors, toothpaste, and microbiological model (*p* = 0.017). No significant interactions were found between toothpaste and incubation time, adherence surface, or test setup, meaning that the toothpastes had the same effect on microbial viability both in EC and BF, independently from the type of bioreactor used. Most importantly, the tested toothpastes had the same effect on microbial viability, both on enamel and RBC surfaces. On the contrary, ANOVA highlighted a significant interaction (*p* < 0.0001) among adherence surface, test setup, microbiological model, and incubation time.

Considering the results of the post-hoc test on the toothpaste factor, the artificial oral microcosm model showed significantly lower EC and BF on group β disks compared to the control, with group α disks showing lower EC and BF compared to control, but higher EC and BF compared to β toothpaste. Surprisingly, this pattern was found on both tested adherence surfaces, demonstrating an influence of toothpastes on EC and BF even over RBC surfaces. In the monospecific *S. mutans* model, the toothpastes did not significantly influence EC and BF.

Considering the role of the adherence surface, the oral microcosm model showed significantly higher EC and BF on enamel than on the tested RBC in both test setups (shaking multiwell plate and MDFR). *S. mutans* showed significantly higher EC and BF on the tested RBC than on enamel in the shaking multiwell plate test setup and significantly higher BF on the tested RBC than on enamel in the MDFR.

Regarding incubation time, as expected, there was an overall significant increase in biomass over time. The artificial oral microcosm grown in the MDFR on enamel surfaces showed no significant difference in viable adherent biomass between 12 and 24 h incubation (*p* = 0.0711). Likewise, *S. mutans* grown in the MDFR on both adherence surfaces showed no significant difference in viable adherent biomass between 12 and 24 h incubation (*p* = 0.5665 and *p* = 0.7691 for enamel and RBC, respectively).

From the test setup point of view, MDFR showed an overall significantly higher EC and BF than the shaking multiwell plate, except for BF of artificial oral microcosm on enamel surfaces.

Considering the microbiological model, the artificial oral microcosm showed higher EC and BF than *S. mutans* on enamel surfaces in both test setups, whereas *S. mutans* showed higher EC and BF than the artificial oral microcosm in the shaking multiwell plate on RBC surfaces.

A set of additional enamel and RBC specimens were made and treated with the toothpastes or with the control, as described in the Methods section, to provide a morphological view of the microbial colonization. The most biologically meaningful result was found to be the one obtained using the MDFR test setup and the artificial oral microcosm model for 12 h (EC, Figure 5). Therefore, this particular situation was furtherly investigated using confocal laser-scanning microscopy and live/dead staining. The results are illustrated in Figure 6. Enamel specimens showed a uniform microbial layer colonizing all surfaces, with multilayered structures starting to develop. The aspect is in keeping with the viable biomass results, showing early colonization of β toothpaste < α toothpaste = control. On enamel specimens treated with β toothpaste, a slightly higher amount of dead cells and microcolonies can be seen. RBC specimens showed much lower colonization than enamel, showing early colonization of β toothpaste < α toothpaste < control. The surfaces of the specimens provided an intense red fluorescence that was not wholly subtracted by parallel scanning in the near-UV 405 nm channel. This artifact was not due to dead cells, but rather to autofluorescence coming from the RBC, and also to an affinity of the propidium iodide dye with RBC surfaces, as previously shown [22].

### 3.3. Fluoride Release Results

The fluoride release from the disks treated with the fluoride-hydroxyapatite containing toothpaste, and the fluoride content of the biofilms after 12 and 24 h are displayed in Figure 7. A significant albeit shallow release of fluoride was seen from enamel specimens treated with β toothpaste. Contrary to RBC specimens, enamel specimens showed biofilms containing significant amounts of fluoride at both incubation times. After 12 h, treatment with β toothpaste significantly decreased fluoride presence in the overlying biofilm compared to the control (*p* < 0.001).

## 4. Discussion

Nano-hydroxyapatite is considered one of the most promising bioactive materials with a broad spectrum of applications in preventive and restorative dentistry. N-HAp particles have been demonstrated to have a similar structure to natural enamel apatite crystals and can remineralize dental hard tissues [15,23,24,25]. Non-substituted n-HAp is also used for desensitizing and anti-erosion purposes [26,27]. The synthesis of nano-scaled zinc carbonate or magnesium-strontium carbonate hydroxyapatite (ZnCO_3_/n-HAp, Mg-SrCO_3_/n-HAp) represented significant progress in this field, allowing the introduction of n-HAp-based toothpastes with remineralization and antibacterial potential [28]. Different substituted n-Hap, coupled to other compounds with antibacterial activity, such as Zn PCA, chitosan, and fluoride, have been developed to achieve the goal of the remineralization of dental hard tissues [4,29,30,31,32]. Nevertheless, very few data are available in the literature about the antibacterial activity of these compounds, which is important in caries prevention because of the impact of oral biofilms in the dynamics of carious lesion formation.

Biofilms produce organic acids that cause the demineralization of the hard tissues. Moreover, biofilms can act as a reservoir of calcium and phosphate for remineralization, but also as a diffusion barrier for other compounds, such as fluoride, thus reducing the efficacy of some caries-preventive agents [16]. From a microbiological point of view, n-HAp-containing products might allow adsorbed nanoparticles to interact with bacterial cells, reducing adherence to hard tissue surfaces, thereby reducing biofilm formation.

The current in vitro study compared the effect of two substituted n-HAp-based toothpastes on early colonization and biofilm formation. Teeth specimens treated with the α toothpaste showed no significant differences in EC and BF compared to the control and β group, whereas specimens treated with β toothpaste showed a significantly lower EC and BF compared to the control (Figure 4 and Figure 5). Both toothpaste formulations contained doping agents having antibacterial activity, namely zinc in α toothpaste and fluoride and strontium in β toothpaste, suggesting that n-HAp doping agents, rather than n-HAp itself, play a crucial role in determining the antibacterial properties of these compounds. Furthermore, fluoride and strontium have been shown to exhibit synergistic antibacterial performance [33,34]. EDS data (Table 2) showed that the zinc and fluoride concentrations were in the same range, while strontium in the β toothpaste was four times higher. A possible explanation of these results could be that the association of fluoride and strontium was more effective in reducing EC and BF than zinc alone at the tested concentrations. Nevertheless, fluoride release (Figure 7), which was only found on brushed enamel surfaces, did not help in explaining the reduced EC and BF by β toothpaste on RBC surfaces. Furthermore, despite a significant initial release of fluoride, the amount of this ion was not increased in biofilms grown over enamel and RBC surfaces treated with the β toothpaste, confirming the fact that biofilms do not tend to accumulate fluoride [16].

EDS data also showed the presence of calcium and phosphate on the surfaces of RBC specimens treated with both toothpastes, as well as traces of zinc on specimens treated with α toothpaste and magnesium on specimens treated with β toothpaste (Figure 2 and Table 1). It is noteworthy that toothpaste remnants could be associated with alumina and barium glass fillers rather than silica particles. This observation, needing further confirmation, opens the possibility for such compounds to actively interact with RBC surfaces with the effect of reducing or preventing secondary caries occurrence.

Regarding the influence of the adherence surface on the antibacterial activity of substituted n-HAp, as the surface features can deeply influence biological behavior, the roughness of all specimens was reduced below 0.2 µm Ra [35] to exclude the possible influence of this parameter on EC and BF. The two tested surfaces were selected considering their buffering capacity: a buffering surface (enamel) and a non-buffering one (RBC). The non-buffering surface was selected because it has been demonstrated to promote the growth of biofilms with a high concentration of cariogenic bacteria [17]. Our data showed that the adherence surfaces did not influence the effect of the tested toothpastes on EC and BF. This result was observed for both *S. mutans* and the oral microcosm, indicating that, unexpectedly, both substituted n-HAps had the same effect on buffering and non-buffering surfaces independent of their chemical composition, suggesting a similar interaction of the active components with the specimen surface. EDS results confirm the presence of active compounds from both toothpastes on both adherence surfaces. This presence was seen to a larger extent on enamel compared to RBC. However, the presence of such active compounds on the tested RBC surface, as seen in the present study, is very interesting from the perspective of controlling EC and BF on these surfaces, with the final aim of reducing secondary caries occurrence. Indeed, small but accumulative effects can prevent dysbiotic changes in dental biofilms and help maintain a beneficial oral microbiome [36]. In this sense, the use of this kind of active principles might be an efficient means of reducing secondary caries occurrence by actively promoting our natural microbiota and reducing the impact of the drivers of dysbiosis, such as, for instance, the presence of a non-buffering surface.

Two test setups were used, shaking multiwell plates and a drip-flow bioreactor to simulate clinical situations as close as possible, avoiding the variabilities that characterize in vivo studies. The first test setup was designed to evaluate EC and BF under low hydrodynamic stress conditions, for instance, during the night. Moreover, the first setup favors a progressive increase in microorganism catabolites and antimicrobial agents released from the surfaces. The second one simulated relatively high hydrodynamic stress conditions that occur during the daytime. *S. mutans* showed lower EC and BF in the first test setup compared to the second one, probably due to the limited amount of nutrients available and the accumulation of metabolic acids in the culture medium.

From the microbiological model point of view, the approach simulated as much as possible the clinical situation by testing an oral microcosm in addition to a monospecific *S. mutans* biofilm. The oral microcosm was used to evaluate the mixed oral flora developing on hard tissues, while the monospecific *S. mutans* model was used to assess the behavior of the cariogenic part of flora when treated with the tested compounds. In our study, *S. mutans* showed a higher affinity for RBC surfaces than for enamel, which is probably related to the buffering capacity of the tested surfaces being a crucial factor for both EC and BF.

Still, the microbiological models and incubation conditions may not be an accurate replica of all the complex interactions taking place in the oral environment. However, in vitro simulations are useful since they can push microenvironmental conditions to levels that might be difficult to control, or even unethical if replicated in vivo. The culture medium was made to have a high content in sucrose, therefore promoting the growth of fermentative Streptococcus as well as the acidogenic species in the mixed inoculum. The aim was to provide a shift in microbial composition towards a pathogenic biofilm, to test the activity of the toothpastes in a harsh environment. An active principle such as substituted n-HAp is especially useful if it can provide a controlled release of ions over time and, especially, in response to defined environmental conditions. A fermentative biofilm can reach pH values lower than 4.0, promoting the demineralization of enamel (critical pH ≈ 5.5). n-HAp is known to be very stable at neutral pH. Therefore, an acidogenic microbial challenge as the one produced in the present setups helped in understanding the effect of the toothpastes when challenged by pathogenic biofilms.

Considering incubation time, 12 h was selected as toothbrushing twice a day is the most frequently advised routine, while a 24 h incubation can provide a sufficient amount of time to evaluate BF. The results showed that the most relevant differences between the tested toothpaste groups were found on *S. mutans* EC, which means after 12 h of incubation. In a translational sense, this may suggest that the tested toothpastes provided the best effect when used twice a day, rather than a single time.

## 5. Conclusions

A toothpaste treatment based on substituted n-HAp showed an in vitro antibacterial effect both on natural and RBC surfaces, which was more pronounced after 12 h than after 24 h. On RBC surfaces, toothpaste remnants seemed to show selective affinity with some particles, such as alumina and barium glass fillers, rather than silica particles. This observation is worthy of future investigations aimed at clarifying the interactions between substituted n-HAp and RBC fillers from the perspective of secondary caries prevention. This phenomenon provides a basis to design clinical studies aimed to confirm in vivo the preventive efficacy of these active principles.

## Figures and Tables

**Figure 1 jfb-11-00036-f001:**
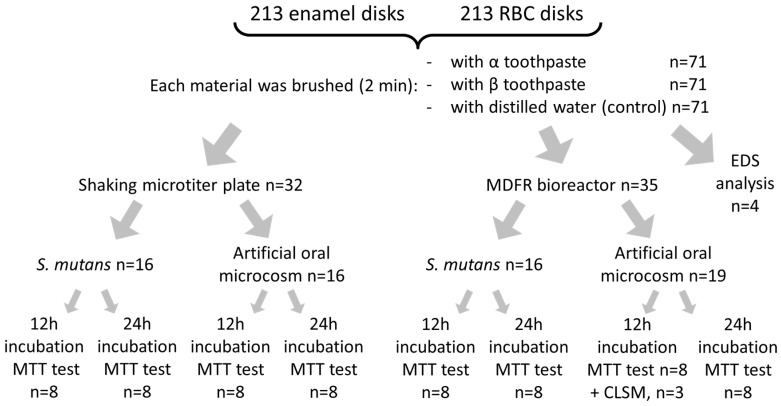
Diagram of specimens processing for the present study. After brushing with the tested toothpastes or with distilled water, enamel and RBC specimens were rinsed, sterilized, then subjected to microbiological analysis and surface imaging (SEM, CLSM) and analysis (EDS). The microbiological analysis included two test setups, two microbiological models, and two incubation times.

**Figure 2 jfb-11-00036-f002:**
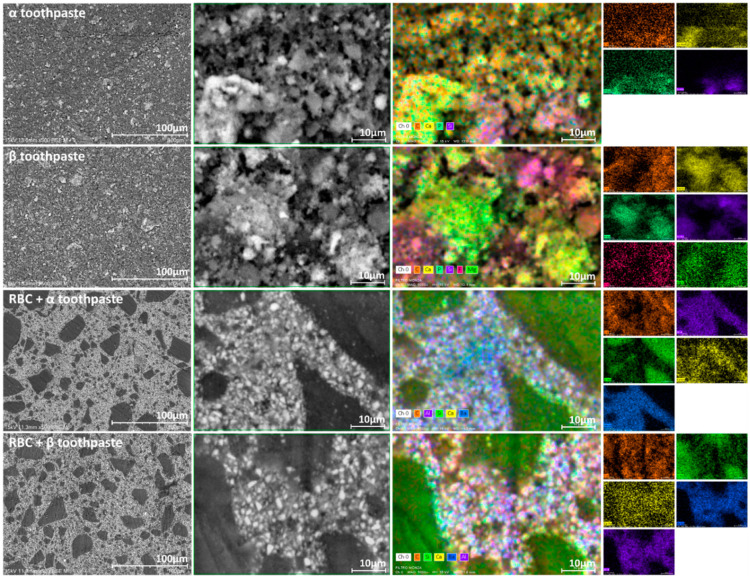
The panel, to be read horizontally, depicts the SEM backscattered electrons micrographs (500× and 5000×, respectively) and EDS elemental maps (5000×) of the specimen surfaces. The EDS maps are additionally presented as single-channel maps to better identify the topographical presence of each element. The first two rows of the panel display the surfaces of the vacuum-dried toothpastes tested in this study: α toothpaste (containing Zn-carbonate substituted n-HAp) and β toothpaste (containing F, Mg, Sr-carbonate substituted n-HAp). Their aspect is very similar, showing silica microparticles (identified by Si signal) and clusters of n-HAp (identified by Ca and P signals). Zn and Sr signals were below the detection limits in mapping mode and were not displayed. The last two rows of the panel represent the surfaces of the tested restorative material after treatment with the toothpastes: RBC + α toothpaste and RBC + β toothpaste. The RBC composition included silica particles (identified by Si signal), and alumina and barium glass micro and nanoparticles (identified by Al and Ba signals). Ca signal was displayed as a marker of toothpaste remnants. It is noteworthy that toothpaste remnants could be associated with alumina and barium glass fillers rather than silica particles.

**Figure 3 jfb-11-00036-f003:**
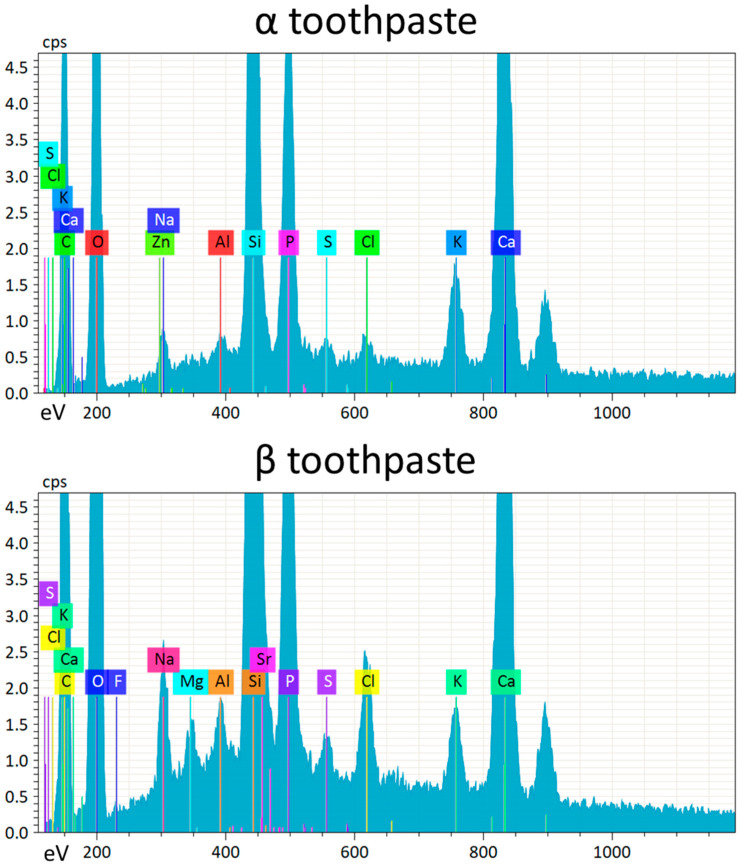
EDS spectra acquired from vacuum-dried tested toothpastes surfaces (α toothpaste containing Zn-carbonate substituted n-HAp, and β toothpaste containing F, Mg, Sr-carbonate substituted n-HAp). Strong Ca and P signals are identified belonging to the n-HAp, as well as the signals corresponding to the corresponding doping elements (Zn in α toothpaste and F, Mg, Sr, in β toothpaste). High counts of Si were also detected in both toothpastes, together with relatively low counts of Al and other elements. The relative amounts of n-HAp doping elements and other elements such as Al and S are below the conventionally considered detection limit of EDS (about 1 wt%). However, the presence of such elements is shown by peaks that were clearly identifiable on all acquired spectra. In this sense, the acquisition of several spectra over the surfaces of many specimens, and the use of statistical analysis on acquired data demonstrating low variability in signals among the different acquisitions (cf. Table 2) allows overcoming, to some extent, this detection limit, improving the performances of EDS analysis.

**Figure 4 jfb-11-00036-f004:**
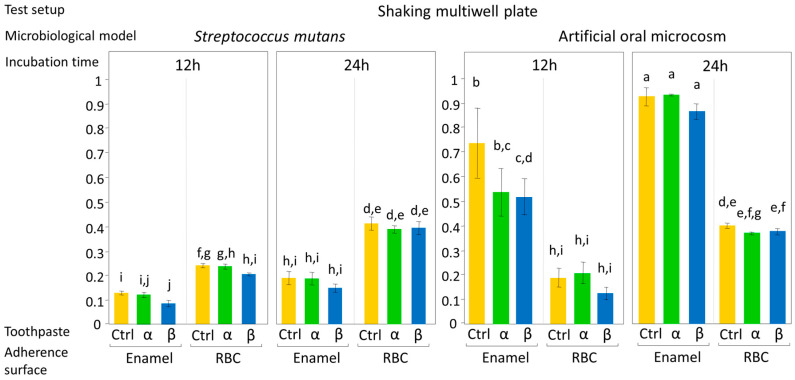
Biofilm formation on the surface of the tested specimens using the shaking multiwell plate test setup, according to the microbiological model (*S. mutans* monospecific biofilm or artificial oral microcosm, aerobically grown) and to the incubation time (12 h or 24 h). Sucrose-enriched sterile modified artificial saliva medium was used in all experiments. Low shear stress on specimens’ surfaces was obtained by an orbital incubator, to simulate oral conditions during the night. Moreover, the closed system setup allows a progressive increase in microorganism catabolites and antimicrobial agents released from the surfaces. Results of viable biomass assay are expressed as mean OD ± SE. Different superscript letters indicate significant differences between groups (student’s test, *p* < 0.05). α toothpaste contains Zn-carbonate substituted n-HAp; β toothpaste contains F, Mg, Sr-carbonate substituted n-HAp while Ctrl group was brushed with distilled water. β toothpaste significantly reduced the early colonization of the artificial oral microcosm on enamel surfaces when compared to the control.

**Figure 5 jfb-11-00036-f005:**
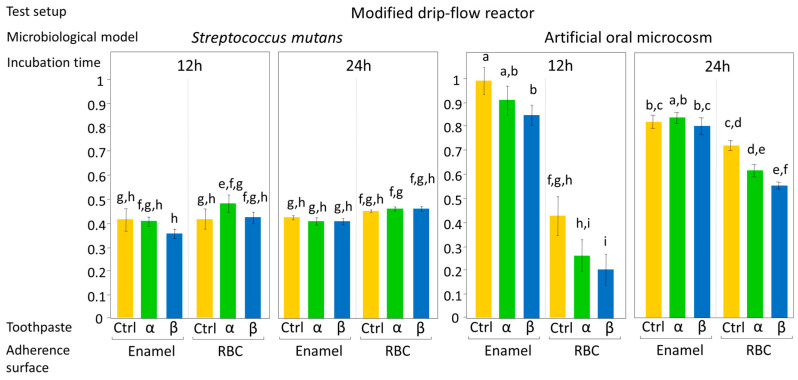
Biofilm formation on the surface of the tested specimens using the MDFR bioreactor test setup, according to the microbiological model (*S. mutans* monospecific biofilm or artificial oral microcosm, aerobically grown) and to the incubation time (12 h or 24 h). Sucrose-enriched sterile modified artificial saliva medium was used in all experiments, being pumped through the flow-cells of the bioreactor. High hydrodynamic stress conditions that occur during the daytime can thus be simulated. An elution of microorganism catabolites and antimicrobial agents released from the surfaces can also occur. Results of viable biomass assay are expressed as mean OD ± SE. Different superscript letters indicate significant differences between groups (student’s test, *p* < 0.05). α toothpaste contains Zn-carbonate substituted n-HAp; β toothpaste contains F, Mg, Sr-carbonate substituted n-HAp while Ctrl group was brushed with distilled water. β toothpaste significantly reduced early colonization (enamel and RBC) and biofilm formation (RBC) of the artificial oral microcosm when compared to the control. Interestingly, the adherence surface showed a more considerable influence than toothpaste treatment on early colonization and biofilm formation, independent of the microbiological model or test setup applied. The effect of the toothpaste on RBC surfaces was not expected and opens the possibility to control microbial colonization on RBCs, and, ultimately, secondary caries prevention, by such treatments.

**Figure 6 jfb-11-00036-f006:**
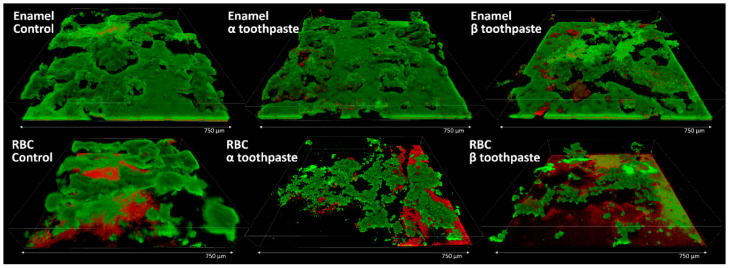
CLSM results of the adherence surfaces treated with the control and the tested toothpastes using the MDFR test setup and the artificial oral microcosm model for 12 h. Scans were analyzed 3D reconstructions obtained using Drishti software. Enamel surfaces provided a much higher early colonization than RBC surfaces. In the background of the enamel control specimen, a central microcolony shows a long tail detached from the surface and oriented horizontally downstream (to the right). This feature is typical of biofilms that develop on surfaces in the presence of relatively high shear stress and is also a means to colonize downstream surfaces rapidly. This feature demonstrates the good morphological resemblance of bioreactor-grown biofilms with in vivo ones. Enamel specimens treated with β toothpaste showed a higher amount of dead cells compared to the other groups. RBC specimens treated with β toothpaste showed the lowest early colonization overall, consistently with viable biomass results.

**Figure 7 jfb-11-00036-f007:**
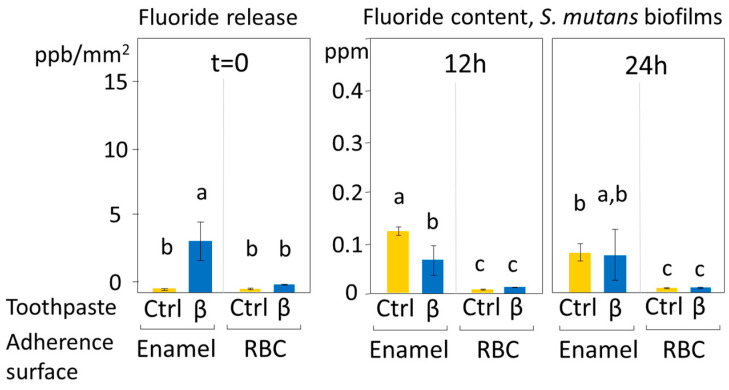
Immediate fluoride release (ppb/mm^2^ ± SE) after disks’ brushing and fluoride presence (ppm ± SE) in the biofilms grown over the disks’ surfaces after 12 and 24 h. Different superscript letters indicate significant differences between groups (Student’s test, *p* < 0.05). β toothpaste contains F, Mg, Sr-carbonate substituted n-HAp while Ctrl group was brushed with distilled water. β toothpaste did not increase the baseline fluoride presence in biofilms grown over enamel surfaces despite showing a low but significant immediate fluoride release. The reduction in fluoride presence seen in enamel specimens after 12 h of incubation when compared to the control may be due to an uptake of fluoride by the tested toothpaste.

**Table 1 jfb-11-00036-t001:** Label, manufacturer and composition of the tested substituted n-Hap-containing toothpastes.

Label	Name	Company	Composition
Toothpaste α	Biorepair Total Protection Plus	Coswell Funo (BO), Italy	Purified water, zinc carbonate hydroxyapatite, glycerin, sorbitol, hydrated silica, silica, aroma, cellulose gum, tetrapotassium pyrophosphate, sodium myristoyl sarcosinate, sodium methyl cocoyl taurate, sodium saccharin, citric acid, phenoxyethanol, benzyl alcohol, sodium benzoate.
Toothpaste β	Biosmalto Caries, Abrasion and Erosion	Curasept Healthcare S.p.A. Saronno (VA), Italy	Purified water, glycerin, hydrated silica, fluoride-hydroxyapatite, Mg-Sr-carbonate hydroxyapatite conjugated with chitosan, cellulose gum, xylitol, cocamidopropyl betaine, xantham gum, aroma, acesulfame K, ethylhexylglycerin, phenoxyethanol, sodium benzoate, citric acid.

**Table 2 jfb-11-00036-t002:** Surface elemental composition of the experimental groups and vacuum-dried toothpastes (α toothpaste containing Zn-carbonate substituted n-HAp and β toothpaste containing F, Mg, Sr-carbonate substituted n-HAp), as assessed by EDS. Data are presented as mean (±1 SD); different letters indicate, for each element, significant differences between groups (Tukey test, *p* < 0.05).

Element	Control Enamel	Control RBC	α Toothpaste	β Toothpaste	Enamel+ α Toothpaste	Enamel+ β Toothpaste	RBC+ α Toothpaste	RBC+ β Toothpaste
C	6.19(0.32) c	33.36(0.49) a	26.08(2.01) b	25.53(1.10) b	5.49(0.75) c	6.12(0.20) c	32.77(1.86) a	33.96(0.34) a
O	38.21(1.38) a,b	33.04(0.33) c,d	43.75(3.23) a	42.69(0.38) a	40.12(2.93) a,b	36.06(1.78) b,c,d	32.89(2.21) d	32.63(0.60) c,d
Ca	38.71(0.90) a	0.00(0.00) c	12.12(2.22) b	9.80(0.40) b	36.97(2.30) a	39.13(0.74) a	1.26(0.12) c	0.30(0.13) c
P	16.31(0.45) a	0.00(0.00) c	4.95(0.95) b	4.93(0.42) b	15.78(1.39) a	16.59(1.20) a	0.76(0.17) c	0.12(0.06) c
Al	0.03(0.04) c	1.46(0.04) a,b	0.29(0.05) b,c	0.32(0.14) b,c	0.09(0.04) c	0.07(0.05) c	1.31(0.78) a,b	1.87(0.03) a
Si	0.08(0.02) d	20.50(0.32) a	9.36(1.81) c	12.12(0.59) b	0.49(0.15) d	0.39(0.04) d	20.23(0.69) a	20.24(0.64) a
Ba	0.00(0.00) c	11.37(0.56) a	0.00(0.00) c	0.00(0.00) c	0.00(0.00) c	0.00(0.00) c	10.52(0.39) b	10.74(0.09) a,b
F	0.00(0.00) b	0.00(0.00) b	0.00(0.00) b	0.18(0.08) a	0.00(0.00) b	0.04(0.02) b	0.00(0.00) b	0.00(0.00) b
Mg	0.18(0.01) b,c	0.00(0.00) d,e	0.00(0.00) d,e	0.35(0.01) a	0.13(0.03) c,d	0.28(0.11) a,b	0.00(0.00) e	0.03(0.01) c,d,e
Zn	0.05(0.06) c	0.00(0.00) c	0.27(0.05) a	0.00(0.00) c	0.17(0.04) a,b	0.05(0.06) c	0.09(0.03) b,c	0.00(0.00) c
Sr	0.00(0.00) c	0.00(0.00) c	0.00(0.00) c	0.93(0.09) a	0.00(0.00) c	0.38(0.10) b	0.00(0.00) c	0.00(0.00) c
Na	0.02(0.03) d	0.00(0.00) d	1.09(0.01) a	1.38(0.01) a	0.52(0.06) b,c	0.71(0.23) b	0.09(0.10) d	0.21(0.04) c,d
S	0.00(0.00) b	0.00(0.00) b	0.25(0.06) a	0.30(0.00) a	0.00(0.00) b	0.00(0.00) b	0.00(0.00) b	0.00(0.00) b
Cl	0.21(0.05) b,c	0.03(0.02) d	0.30(0.07) b	1.08(0.13) a	0.24(0.03) b	0.22(0.01) b,c	0.07(0.01) d	0.09(0.02) c,d
K	0.00(0.00) c	0.00(0.00) c	1.56(0.02) a	0.92(0.06) b	0.00(0.00) c	0.00(0.00) c	0.07(00.09) c	0.02(0.00) c

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
