# Peer review of "Substituted Nano-Hydroxyapatite Toothpastes Reduce Biofilm Formation on Enamel and Resin-Based Composite Surfaces"

_jfb, 2020, doi:10.3390/jfb11020036_

Round 1

Reviewer 1 Report

The paper describes a comparison of two commercial toothpastes that contain nano-hydroxyapatite (n-HAp) substituted with metal ions. The effect of these two commercial products is compared with the use of distilled water, using different materials, methods of incubation and biofilms (S. mutans o mixed biofilm). The results are interesting, although the work would have increased its interest by checking different incubation conditions (anaerobic?), analysing the effect of the compounds on the species composition of the mixed biofilm by metagenomic techniques and comparing the effect of the commercial toothpastes with more appropriate controls (i.e. pure n-HAp, tooth paste without n-HAp..).

Still, the paper may deserve publication, but some changes should be introduced:

I think the numbers of the figures are all mixed up in the text. I.e., in lines 82-88: Do you refer to Table 1? I cannot see any link to Figure 2 or Table 2. Please check thoroughly. I think that the Figure numbers are all changed.

The results of SEM analysis (Figure 1) should be explained in the results, not in the methods, and probably the figure is more suitable for Supplementary materials.

In the discussion you should consider that culture medium and incubation conditions may have altered the results. Your culture medium has a high content in sucrose, and therefore promotes the growth of fermentative Streptococcus. This should be discussed

I understand the difficulties in selecting a suitable toothpaste control other than water, but this fact should be mentioned in the discussion

Specific comments:

In the abstract and throughout the text: Chante “Oral microflora” by “oral microbiota”

Abstract: line 30. The expression “intermediate values” is not clear. The same sentence is present in the discussion.

Please state in the abstract that you are using commercial products

Line 33. I cannot see any paragraph in the discussion supporting the conclusion of the “effect on secondary caries”

Terminology for the toothpastes is very confusing. It is OK for the blind experimental design, but it is little informative for the paper. Gamma (water) treatment should be labelled as “control” or “water”. Generally, the control is represented first in the tables/figures. Please state the meaning of Alpha, beta and gamma in the figure legends. While description of the results is not required in the figure legends, it would be desirable that you include some other information regarding culture conditions (aerobiosis/anaerobiosis, culture medium, etc)

Methods: Please state if plates were incubated in aerobic or anaerobic conditions

Author Response

Response to Reviewer 1 Comments

The paper describes a comparison of two commercial toothpastes that contain nano-hydroxyapatite (n-HAp) substituted with metal ions. The effect of these two commercial products is compared with the use of distilled water, using different materials, methods of incubation and biofilms (S. mutans o mixed biofilm). The results are interesting, although the work would have increased its interest by checking different incubation conditions (anaerobic?), analysing the effect of the compounds on the species composition of the mixed biofilm by metagenomic techniques and comparing the effect of the commercial toothpastes with more appropriate controls (i.e. pure n-HAp, tooth paste without n-HAp..).

Still, the paper may deserve publication, but some changes should be introduced:

Point 1: I think the numbers of the figures are all mixed up in the text. I.e., in lines 82-88: Do you refer to Table 1? I cannot see any link to Figure 2 or Table 2. Please check thoroughly. I think that the Figure numbers are all changed.

Response 1: The Reviewer is right, we apologize for this mishap. The numbering of figures and tables has been amended throughout the manuscript.

Point 2: The results of SEM analysis (Figure 1) should be explained in the results, not in the methods, and probably the figure is more suitable for Supplementary materials.

Response 2: We added, as suggested by the Reviewer, the following sentence to the results: “To investigate the presence of toothpaste residues on enamel and RBC surfaces as well as the composition of the tested toothpastes,  Specimens were observed using SEM-EDS in surface-charge reduction mode without sputter-coating, using an accelerating voltage of 15 KV (Figure 1).” We kindly disagree with moving the Figure into supplementary material, since the results it displays are crucial to our findings, that there is an effect of the toothpastes even on RBC surfaces. This was not put into evidence before.

Point 3: In the discussion you should consider that culture medium and incubation conditions may have altered the results. Your culture medium has a high content in sucrose, and therefore promotes the growth of fermentative Streptococcus. This should be discussed

Response 3: We thank the Reviewer for addressing this point. The aim of this study was to test the effect of the n-HAp toothpastes under cariogenic conditions, not under “normal” clinical conditions, but this was not specified nor discussed. The following paragraph has been added to the discussion:

“Still, the microbiological models and incubation conditions may not be an accurate replica of all the complex interactions taking place in the oral environment. However, in vitro simulations are also useful since they can push microenvironmental conditions to levels that might be difficult to control, or even unethical if replicated in vivo. The culture medium was made to have a high content in sucrose, therefore promoting the growth of fermentative Streptococcus as well as the acidogenic species in the mixed inoculum. The aim was to provide a shift in microbial composition towards a pathogenic biofilm, to test the activity of the toothpastes in a harsh environment. An active principle such as substituted n-HAp is especially useful if it can provide a controlled release of ions over time and, especially, in response to defined environmental conditions. A fermentative biofilm can reach pH values lower than 4.0, promoting the demineralization of enamel (critical pH ≈ 5.5). n-HAp is known to be very stable at neutral pH. Therefore, an acidogenic microbial challenge as the one produced in the present setups helped in understanding the effect of the toothpastes when challenged by pathogenic biofilms.”

Point 4: I understand the difficulties in selecting a suitable toothpaste control other than water, but this fact should be mentioned in the discussion

Response 4: We definitely agree with the Reviewer, and the study was originally designed to have that control. When we performed the tests, we had a fourth treatment group made by a control, silica-based toothpaste without n-HAp, manufactured by one of the companies that produced the toothpastes we tested. Problem is, they put into that formulation many other unrequested compounds, including preservatives in high concentration, jeopardizing the role of a control toothpaste especially in a microbiological study. Being a blind study, we noticed that only at the end of the experiments and we were forced, therefore, to exclude that group from the analysis.

Specific comments:

Point 5: In the abstract and throughout the text: Chante “Oral microflora” by “oral microbiota”

Response 5: This was done. Thank you.

Point 6: Abstract: line 30. The expression “intermediate values” is not clear. The same sentence is present in the discussion.

Response 6: That expression was substituted with a more detailed description: “α toothpaste showed lower EC and BF compared to control, but higher EC and BF compared to β toothpaste”.

Point 7: Please state in the abstract that you are using commercial products

Response 7: This was done. Thank you.

Point 8: Line 33. I cannot see any paragraph in the discussion supporting the conclusion of the “effect on secondary caries”

Response 8: The Reviewer is right, we added to the introduction a better explanatory paragraph, as follows: “Resin-based composites (RBCs) provide an example of artificial surfaces intensely colonized by cariogenic biofilms, due to their lack of buffering effect. This characteristic has been put into relation with the high occurrence of secondary caries in the sound dental tissues surrounding an RBC restoration. The possibility of influencing microbial colonization of RBCs, therefore, can be seen as a means to prevent the occurrence of such lesions. For this reason, RBCs are a potential target of preventive measures exploiting the potential of nanotechnologies in order to reduce secondary caries occurrence.”

The following sentences were also added to the discussion: “The presence of such active compounds on the tested RBC surface, as seen in the present study, is very interesting from the perspective of controlling EC and BF on these surfaces, with the final aim of reducing secondary caries occurrence. Indeed, small but accumulative effects can prevent dysbiotic changes in dental biofilms and help maintain a beneficial oral microbiome [36]. In this sense, the use of this kind of active principles might be an efficient means of reducing secondary caries occurrence by actively promoting our natural microbiota and reducing the impact of the drivers of dysbiosis, such as, for instance, the presence of a non-buffering surface."

Point 9: Terminology for the toothpastes is very confusing. It is OK for the blind experimental design, but it is little informative for the paper. Gamma (water) treatment should be labelled as “control” or “water”. Generally, the control is represented first in the tables/figures. Please state the meaning of Alpha, beta and gamma in the figure legends.

Response 9: We agree with the Reviewer. Gamma treatment was labelled everywhere as “control”. It was represented first in all figures and tables. The active principles of alpha and beta toothpastes were specified in all figure legends. Thank you.

Point 10: While description of the results is not required in the figure legends, it would be desirable that you include some other information regarding culture conditions (aerobiosis/anaerobiosis, culture medium, etc)

Response 10: The following information were included:

Figure 3: Sucrose-enriched sterile modified artificial saliva medium was used in all experiments. Low shear stress on specimens’ surfaces was obtained by an orbital incubator, to simulate oral conditions during the night. Moreover, the closed system setup allows a progressive increase in microorganism catabolites and antimicrobial agents released from the surfaces.

Figure 4: Sucrose-enriched sterile modified artificial saliva medium was used in all experiments, being pumped through the flow-cells of the bioreactor. High hydrodynamic stress conditions that occur during the daytime can thus be simulated. An elution of microorganism catabolites and antimicrobial agents released from the surfaces can also occur.

Point 11: Methods: Please state if plates were incubated in aerobic or anaerobic conditions

Response 11: This was done. Thank you.

Reviewer 2 Report

This article is well-written and the experimental set-up and methodology is mainly suitable. The article is quite difficult to navigate due to the lack of figure labels and links between the text and data. The main conclusion is that these toothpastes show higher antimicrobial activity after 12h than 24h and brushing twice a day is recommended.. While is this is not a surprising result, the presentation of the data and statistical analysis could be improved to support this conclusion. Indeed, the modified drip-flow reactor shows little different in microbial activity on enamel surfaces. Further discussion is warranted. The other conclusions regarding particle affinity were. not the focus of this study and have very little data to support this conclusion. This should be removed or further data added.

Figure 1. Some scale bars are difficult to see in Figure 1, please make this clearer or add if missing. The figure caption is too lengthy and describes/ discusses results and this should be moved to the main text. Captions should be used to identify the the contents of each figure panel. The figures are rich in data with separate panels and need to be labelled so it is clear what is being referred to the main text and in each caption. Figures are not referred to in the discussion section which makes it difficult to follow and interpret.

The n-HAp substituted ions for toothpastes alpha and beta are below the detection limit for EDS analysis and I suggest this is not the most appropriate technique to use. Could a different technique be used - eg. ToF-SIMS or XPS?

Figure 2. Label each part and show the statistical analysis and the degree of significance between each dataset where commented upon in the discussion. 

Figure 3. same comments as for Figure 2.

Author Response

This article is well-written and the experimental set-up and methodology is mainly suitable.

Point 1: The article is quite difficult to navigate due to the lack of figure labels and links between the text and data.

Response 1: The Reviewer is right, and all this has been improved. See also the answers to Reviewer 1’s comments.

Point 2: The main conclusion is that these toothpastes show higher antimicrobial activity after 12h than 24h and brushing twice a day is recommended. While is this is not a surprising result, the presentation of the data and statistical analysis could be improved to support this conclusion. Indeed, the modified drip-flow reactor shows little different in microbial activity on enamel surfaces. Further discussion is warranted. The other conclusions regarding particle affinity were. not the focus of this study and have very little data to support this conclusion. This should be removed or further data added.

Response 2: We are sorry that the Reviewer did not see what is in our opinion the main result of this study: the fact that the tested toothpastes had an effect on EC and BF even on RBC surfaces. This is crucial for many reasons. Toothpastes and other dental hygiene products are currently not produced to elicit a response from the artificial materials that may be present in a patient’s mouth. Our study demonstrates that it is possible for toothpastes to interact with restorative materials’ surfaces, and the conclusions regarding particle affinity provide a possible explanation for this interesting and previously unknown fact. Since this interaction produces a decrease in EC and BF, this can be a very interesting possibility to reduce secondary caries occurrence, which is the main reason for RBC failure nowadays. The manuscript has been modified so that this point of view emerges more clearly. Finally, it is not necessary to obtain huge reductions in EC and BF values, since small but accumulative changes, such as the ones that a subject can perform by repeatedly brush his or her teeths with a n-HAp containing toothpaste, can put a significant pressure on biofilms towards eubiosis. To this regard, the paper by Phil Marsh has been added to the discussion. Marsh, P. D. In sickness and in health—what does the oral microbiome mean to us? An ecological perspective. Adv. Dent. Res. 2018, 29(1), 60-65.

Point 3: Figure 1. Some scale bars are difficult to see in Figure 1, please make this clearer or add if missing. The figure caption is too lengthy and describes/ discusses results and this should be moved to the main text. Captions should be used to identify the the contents of each figure panel. The figures are rich in data with separate panels and need to be labelled so it is clear what is being referred to the main text and in each caption. Figures are not referred to in the discussion section which makes it difficult to follow and interpret.

Response 3: Scale bars were added more clearly. The last sentence of the figure caption was moved to the main text to shorten it. Figures were labelled and captions provided.

Point 4: The n-HAp substituted ions for toothpastes alpha and beta are below the detection limit for EDS analysis and I suggest this is not the most appropriate technique to use. Could a different technique be used - eg. ToF-SIMS or XPS?

Response 4: We agree with the Reviewer, yes-of course these techniques usually do provide better results especially where low amounts of active principles are expected. However, there is not a unique threshold of detection limit for all EDS probes, since, especially modern high-resolution cooled sensors are much more sensitive than older systems where a 1-3% detection limit was to be expected. We added two of the acquired spectra in Figure 2 to show that even low amounts of elements can be traced and identified after careful manual settings, and increasing the observation area and the acquisition time. Please also note that many times EDS is used as a qualitative, not quantitative imaging analysis so researchers most often do not acquire many fields of the same group to perform statistical analysis on elemental dataset.

The added legend reads as follows:

EDS spectra acquired from vacuum-dried tested toothpastes surfaces (α toothpaste containing Zn-carbonate substituted n-HAp, and β toothpaste containing F, Mg, Sr-carbonate substituted n-HAp). Strong Ca and P signals are identified belonging to the n-HAp, as well as the signals corresponding to the corresponding doping elements (Zn in α toothpaste and F, Mg, Sr, in β toothpaste). High counts of Si were also detected in both toothpastes, together with relatively low counts of Al and other elements. The relative amounts of n-HAp doping elements and other elements such as Al and S are below the conventionally considered detection limit of EDS (about 1 wt%). However, the presence of such elements is shown by peaks that were clearly identifiable on all acquired spectra. In this sense, the acquisition of several spectra over the surfaces of many specimens, and the use of statistical analysis on acquired data demonstrating low variability in signals among the different acquisitions (cf. Table 1) allows overcoming, to some extent, this detection limit, improving the performances of EDS analysis.

Point 5: Figure 2. Label each part and show the statistical analysis and the degree of significance between each dataset where commented upon in the discussion.

Figure 3. same comments as for Figure 2.

Response 5: The Figure legends have been changed to better explain the different parts of the graph and to show the statistical analysis.

Thank you.

Reviewer 3 Report

The authors have proposed that substituted nano-hydroxyapatite toothpastes reduces biofilm formation on enamel and resin-based composite surfaces. 

However, results related with reduced biofilm formation are very weak. 

There are other methods to verify the biofilm formation such as confocal imaging or crystal violet staining. 

MTT assay does not directly reflect biofilm formation but rather reflect the number of viable bacteria in the specimen. 

Viable cell count should be included to support MTT result. 

Author Response

The authors have proposed that substituted nano-hydroxyapatite toothpastes reduces biofilm formation on enamel and resin-based composite surfaces.

Point 1: However, results related with reduced biofilm formation are very weak.

Response 1: We agree with the Reviewer, however, as pointed out in answer to R2, it is not necessary to obtain considerable reductions in EC and BF values, since small but accumulative changes, such as the ones that a subject can perform by repeatedly brushing his or her teeth with an n-HAp containing toothpaste, can put significant pressure on biofilms towards eubiosis. In this regard, the paper by Phil Marsh has been added to the discussion. Marsh, P. D. In sickness and in health—what does the oral microbiome mean to us? An ecological perspective. Adv. Dent. Res. 2018, 29(1), 60-65.

Point 2: There are other methods to verify the biofilm formation such as confocal imaging or crystal violet staining.

Response 2: Both alternative methods suggested by the Reviewer have huge limitations, especially in the case of the experiments performed in the present study. CLSM is a very useful tool to study the macromorphology of microbial colonies, especially when organized in biofilms. However, subtle changes in biofilm viability or its metabolic activity, albeit important, are often impossible to be detected using CLSM, or could be excessively time-consuming if a real quantitative approach is wanted. A quantitative biochemical test is preferable. The crystal violet staining is a technique that was widely used in the past, and some still use it today. Its main drawbacks depend on the fact that it stains everything: dead cells, live cells, debris, EPS matrix, not to say about specimens’ surfaces, especially when artificial ones such as RBC are employed. The results of this test are, therefore, always to be interpreted with extreme caution.

Point 3: MTT assay does not directly reflect biofilm formation but rather reflect the number of viable bacteria in the specimen.

Response 3: We agree with the Reviewer, and that was precisely the reason we did it! MTT assay, as we performed it, allows quantifying the number of viable and metabolically active cells. We know that n-HAp is not antibacterial, but the doping ions, especially Zn, Sr, and F are highly biocidal. However, their concentration is not sufficient to achieve biocidal activity, due to the elution and to the fact that biofilms tend to actively get rid of them. However, bacteria can sense the presence of these ions and react by tuning down their own metabolism, which in the long term, also results in a lower amount of viable bacterial cells. For this reason, biochemical tests that are based on tetrazolium salts, or resazurin, or the measurement of ATP level inside the cell are the ideal choice. Please, also refer to Berridge, Michael V., Patries M. Herst, and An S. Tan. "Tetrazolium dyes as tools in cell biology: new insights into their cellular reduction." Biotechnology annual review 11 (2005): 127-152.

Point 4: Viable cell count should be included to support MTT result.

Response 4: In the range from 0.1 to about 1.5 OD, there is a linear relationship between CFU counts and MTT results, provided that possible confounding parameters are kept under control. This has been demonstrated over time many times by many research groups, including ours. While this is certainly valid for S. mutans, it will surely not be the same for the artificial oral microcosm, where one cannot identify by plating most of the microorganisms that constitute the oral microbiota. In this case, plate counts will produce results with high variability and limited statistical significance and will only be based on plate-grown strains, which does not accurately reproduce the microbiota diversity. These results, therefore, would have little significance.

Please, also refer to Sylvester, Paul W. "Optimization of the tetrazolium dye (MTT) colorimetric assay for cellular growth and viability." Drug design and discovery. Humana Press, 2011. 157-168.

Thank you.

Round 2

Reviewer 2 Report

The authors have added clarification where requested and the overall presentation of the manuscript is improved. Thank you

Author Response

The authors have added clarification where requested and the overall presentation of the manuscript is improved. Thank you

Response: We wish to thank you for giving us the opportunity to improve it!

Reviewer 3 Report

The authors have proposed that substituted nano-hydroxyapatite toothpastes reduces biofilm formation on enamel and resin-based composite surfaces.

A biofilm comprises a consortium of microorganisms in which cells stick to each other and often also to a surface. These adherent cells become embedded within a slimy extracellular matrix that is composed of extracellular polymeric substances.

however, there are no direct evidence that the bacteria used in this study directly formed biofilm during the experiment. As the authors have responded. MTT assay can only reflect viable cells that have attached to the surface. 

Authors should provide direct evidence of biofilm formation in the data to say that the tested toothpastes reduced biofilm formation. Other wise, data in the study can only suggest that the tested toothpaste has influenced bacterial adhesion and proliferation on the surface.  

Authors have responded that in the range from 0.1 to about 1.5 OD, there is a linear relationship between CFU counts and MTT results. It would be very helpful if % difference data could be provided. Although there are many significant results compared between groups, which result do the authors suggest to be most biologically meaning?    

Author Response

We are grateful to the Reviewer for the additional suggestions.

In fact, we had started working on obtaining some CLSM images already after reading the first review to provide direct evidence that the bacteria used in this study directly formed biofilm during the experiment and, due to time constraints and to all the previously mentioned considerations, we decided to focus on the most biologically meaningful results. The following parts were therefore added to the results section, together with Figure 6, that shows the CLSM reconstructions.

To provide a morphological view of the microbial colonization and biofilm formation, a set of additional specimens of enamel and RBC were made and treated with the toothpastes or with the control, as described in the Methods. The most biologically meaningful result was found to be the one obtained using the MDFR test setup and the artificial oral microcosm model for 12 h (EC, Figure 4). Therefore, this particular situation was furtherly investigated using confocal laser-scanning microscopy and live/dead staining. The results are illustrated in Figure 5. Enamel specimens showed a uniform microbial layer colonizing all surfaces, with multilayered structures starting to develop. The aspect is in keeping with the viable biomass results, showing early colonization of β toothpaste < α toothpaste = control. On enamel specimens treated with β toothpaste, a slightly higher amount of dead cells and microcolonies can be seen. RBC specimens showed much lower colonization than enamel, showing early colonization of β toothpaste < α toothpaste < control. The surfaces of the specimens provided an intense red fluorescence that was not wholly subtracted by parallel scanning in the near-UV 405 nm channel. This artifact was not due to dead cells, but rather to autofluorescence coming from the RBC, and also to an affinity of the propidium iodide dye with RBC surfaces, as previously shown [22].

Figure 5. CLSM results of the adherence surfaces treated with the control and the tested toothpastes using the MDFR test setup and the artificial oral microcosm model for 12 h. Scans were analyzed 3D reconstructions obtained using Drishti software. Enamel surfaces provided a much higher early colonization than RBC surfaces. In the background of the enamel control specimen, a central microcolony shows a long tail detached from the surface and oriented horizontally downstream (to the right). This feature is typical of biofilms that develop on surfaces in the presence of relatively high shear stress and is also a means to colonize downstream surfaces rapidly. This feature demonstrates the good morphological resemblance of bioreactor-grown biofilms with in vivo ones. Enamel specimens treated with β toothpaste showed a higher amount of dead cells compared to the other groups. RBC specimens treated with β toothpaste showed the lowest early colonization overall, consistently with viable biomass results.

Materials and methods:

4.7. Confocal laser-scanning microscopy (CLSM)

A total of three specimens for each material and treatment group were additionally prepared for CLSM analysis. They were incubated using the MDFR test setup with the artificial oral microcosm model for 12 h. Then, they were gently removed from the flow cells, rinsed twice with sterile PBS, stained using the FilmTracerTM LIVE/DEAD® Biofilm Viability Kit (Invitrogen Ltd., Paisley, UK), and analyzed using confocal laser-scanning microscopy (CLSM; Eclipse Ti2 inverted CLSM, Nikon, Tokyo, Japan). Three randomly selected image stack sections were recorded for each specimen. Confocal images were obtained using a dry objective (20×; NA = 0.5) at a resolution of 2048 × 2048 pixels, with a zoom factor of 1.0 and a scan speed of 400 Hz and digitalized using the Nikon Imaging Software (NIS) – Elements Viewer, v. 4.50. Three channels were acquired in parallel scanning; the first used an excitation at 405 nm and emission at 420–470 nm in order to subtract potential autofluorescence digitally. The other two channels had an excitation wavelength of 488 nm, and emission was acquired at 500–570 nm (green channel, live bacteria) and 610–760 nm (red channel, dead bacteria). For each image stack section, 3D-rendering reconstructions were obtained using Drishti (Ajay Limaye, Australian National University, CAN, AUS, http://sf.anu.edu.au/Vizlab/drishti/).

Authors do believe that providing % difference data would further decrease the readability and understanding of an already very complex study.